# Hemolymph Ecdysteroid Titer Affects Maternal mRNAs during *Bombyx mori* Oogenesis

**DOI:** 10.3390/insects12110969

**Published:** 2021-10-27

**Authors:** Meirong Zhang, Pingzhen Xu, Tao Chen

**Affiliations:** 1Jiangsu Key Laboratory of Sericultural Biology and Biotechnology, School of Biotechnology, Jiangsu University of Science and Technology, Zhenjiang 212028, China; xpz198249@just.edu.cn; 2Key Laboratory of Silkworm and Mulberry Genetic Improvement, Ministry of Agriculture, Sericultural Research Institute, Chinese Academy of Agricultural Sciences, Zhenjiang 212028, China

**Keywords:** *Bombyx mori*, maternal mRNAs, 20-hydroxyecdysone, pupa, wandering

## Abstract

**Simple Summary:**

Both maternal genes and ecdysteroids play important roles during embryonic development. In this study, we aimed to characterize the dynamic landscape of maternal mRNAs and the relationship between maternal genes and ecdysteroids during silkworm oogenesis. For the first time, we determined the start of the accumulation of maternal mRNAs in the ovary at the wandering stage during the larval period. We detected the developmental expression profiles of each gene in the ovary or ovariole. We finally confirmed the role of 20-hydroxyecdysone in regulating maternal gene expression. Taken together, our findings expand the understanding of insect oogenesis and provide a perspective on the embryonic development of the silkworm.

**Abstract:**

Silkworm larval–pupal metamorphosis and the first half of pupal–adult development occur during oogenesis from previtellogenesis to vitellogenesis and include two peaks of the hemolymph ecdysteroid titer. Moreover, a rise in 20-hydroxyecdysone titer in early pupae can trigger the first major transition from previtellogenesis to vitellogenesis in silkworm oogenesis. In this study, we first investigated the expression patterns of 66 maternal genes in the ovary at the wandering stage. We then examined the developmental expression profiles in six time-series samples of ovaries or ovarioles by reverse transcription–quantitative PCR. We found that the transcripts of 22 maternal genes were regulated by 20-hydroxyecdysone in the isolated abdomens of the pupae following a single injection of 20-hydroxyecdysone. This study is the first to determine the relationship between 20-hydroxyecdysone and maternal genes during silkworm oogenesis. These findings provide a basis for further research into the embryonic development of *Bombyx mori*.

## 1. Introduction

Early embryonic development occurs in the absence of de novo transcription and is maternally regulated [1,2]. Maternal mRNAs and proteins stored in the egg during oogenesis are activated to initiate and regulate embryonic development [2,3,4]. Maternal mRNAs are responsible for early embryogenesis by driving cellular division until zygotic genome activation [2]. The handoff of developmental control from the maternal to the zygotic genome is known as the maternal to zygotic transition (MZT). Most striking during the MZT is the elimination of maternal transcripts and the onset of transcription from the newly activated zygote’s genome [2,5,6]. This transition has been well studied in model organisms, and many early developmental processes are highly conserved and are critical for organism survival [7]. RNA-binding proteins (RBPs) regulate the translation, stability, and localization of maternal mRNAs [8]; Smaug (SMG) RBP is essential to maternal mRNAs degradation in *Drosophila*
*melanogaster* [9]. Another type of RBP comprises the AU-rich element-binding proteins (ARE-BPs), which have a role in maternal mRNA clearance during the MZT in *Caenorhabditis elegans* [10], zebrafish (*Danio rerio*) [11], *Xenopus laevis* [12], and mice (*Mus musculus*) [13]. The transcription factors critical to the activation of the zygotic genome following maternal mRNA degradation have been identified in *Drosophila* (Zelda) [14], zebrafish (Nanog, Pou5f1 and SoxB1) [15], mice (Dux, Nfy, Dppa2, and Dppa4) [16,17,18], and humans (OCT4 and DUX4) [16,19,20].

The domestic silkworm, *Bombyx mori* (*B*. *mori*), is not only economically prized for its silk production, but is also as a model of Lepidoptera [21,22]. Therefore, it is widely used in basic and applied research. The ovary structure of the silkworm is polytrophic meroistic. In silkworms, oogenesis can be subdivided into the 3 distinct developmental periods of previtellogenesis, vitellogenesis, and choriogenesis, while the development of the follicles is divided into 12 different stages according to various morphological criteria [23]. The oogenesis developmental period of previtellogenesis consists of stages 1–3 during the larval period. After larval–pupal ecdysis, the ovarioles emerge gradually via elongation into the abdominal cavity, which is part of the oogenesis developmental periods of vitellogenesis and choriogenesis [23].

Throughout the latter half of the fifth instar, a small rise in ecdysteroid titer occurs upon commencement of wandering, and this is followed by a plateau. The next day, the titer starts to increase gradually and then elevates steeply to form a peak 1 day later [24]. After this peak, the ecdysteroid titer decreases rapidly to a very low level, but then increases again after pupation up to a very high titer (forming the second peak) 2 days after pupation. The titer then decreases irregularly to reach a minimum at 6 days after pupation [24]. There is a strong parallel between the overall changes in follicular development and the ecdysteroid titers during larval–pupal and the first half of pupal–adult development. The increase in 20-hydroxyecdysone (20E) titer in the early pupa stage can trigger previtellogenic development and vitellogenesis in the silkworm [25,26,27]. The ecdysteroids in silkworm ovaries are synthesized in follicle cells and transferred into the oocyte where they are phosphorylated by EcKinase, the mRNA of which originates from nurse cells and the oocyte itself [28,29].

The silkworm ovariole has unique features in the context of physiological and biochemical studies, which provide an excellent model for studies on changes in gene expression [30,31,32]. We previously reported that 66 maternal genes have been successfully identified via orthologous comparison and expression detection, and their mRNAs form three clusters of degradation patterns during the MZT period in the silkworm [32,33]. Although the roles of maternal genes and ecdysteroids in insect embryonic development are well known, their relationships remain to be determined. Therefore, in the present study, we first investigated the expression characteristics of 66 maternal genes in the ovary of the wandering silkworm and then examined their developmental expression profiles in the ovaries or ovarioles of silkworms at stages 3 to 5 of oogenesis by reverse transcription–quantitative PCR (RT-qPCR). We found that the transcripts of 22 maternal genes were regulated in the isolated abdomens of the pupae by a single injection of 20E. Our data also demonstrate the start of the accumulation of maternal mRNAs during stage 3 of the oogenesis developmental period of previtellogenesis in the larva.

## 2. Materials and Methods

### 2.1. Experimental Animals and 20-Hydroxyecdysone

*B*. *mori* (Dazao) larvae were reared under standard conditions (25 °C and 70% humidity). The larvae–pupae, pupae, and ligation pupae were maintained under a 12 h light/12 h dark photoperiod at 25 °C and 70% humidity. 20-hydroxyecdysone (20E), an ecdysteroid (A506554), was purchased from Sangon Biotech Co., Ltd. (Shanghai, China) and was dissolved in anhydrous ethanol and diluted to 2 g L^–1^ using sterile distilled water.

### 2.2. Ligature and 20-Hydroxyecdysone Treatment

Immediately after pupation, female pupae of similar sizes were ligated between the thorax and the abdomen using cotton thread. The isolated abdomens of the pupae (20E-depleted abdomens) were treated with 5 μL 20E (10.0 μg) of solution or 5 μL of sterile distilled water (control) four days later.

### 2.3. Sample Preparation

The ovaries or ovarioles were collected from the wandering larvae, pre-pupae, day-0 pupae, day-1 pupae, day-2 pupae, and day-3 pupae and were injected with 20E at 24 h and 48 h after depletion of the abdomens. The dissection was conducted on ice for purposes of anaesthetization.

### 2.4. Reverse-Transcription PCR and Reverse Transcription–Quantitative PCR Analysis

Total RNA was extracted using TRIzol reagent (Invitrogen, Carlsbad, CA, USA). The total RNA concentrations were quantified, and a fraction of the RNA was treated with DNase. The RNA was used to synthesize first-strand cDNA using the PrimeScript™ RT Master Mix (Perfect Real Time; TaKaRa, Dalian, China) according to the manufacturer’s instructions. The expression patterns of maternal genes in the ovaries of the wandering larvae were analyzed using reverse-transcription PCR (RT-PCR) and reverse-transcription–quantitative PCR (RT-qPCR). The ovaries or ovarioles were collected from the wandering larvae, pre-pupae, day-0 pupae, day-1 pupae, day-2 pupae, and day-3 pupae and were used to determine the developmental expression profiles of each gene via RT-qPCR. The ovarioles of pupae injected with 20E at 24 h and 48 h after depletion of the abdomens were used for maternal gene RT-qPCR analysis. The thermal cycling conditions of reverse-transcription PCR (RT-PCR) were 94 °C for 5 min, 35 cycles at 94 °C for 30 s, 58 °C for 30 s, 72 °C for 30 s, and a final extension at 72 °C for 10 min before storing at 12 °C. The RT-PCR products of each gene were separated by 1.2% agarose gel electrophoresis. RT-qPCR was carried out in an ABI PRISM^®^ 7500 (Applied Biosystems, USA) using SYBR Green Supermix (TaKaRa, China). The thermal program of RT-qPCR consisted of an initial denaturation at 95 °C for 3 min, 40 cycles at 95 °C for 15 s, and 60 °C for 31 s, and melting from 60 °C to 95 °C. Translation initiation factor 4a (*TIF*-*4A*) was used as a reference gene [34]. The cycle threshold (Ct) values were converted to linear values using the comparative CT method [35]. Student’s *t*-tests were used to analyze the data. The specific primers for each gene are shown in Appendix A.

### 2.5. Kyoto Encyclopedia of Genes and Genomes Annotation

Kyoto Encyclopedia of Genes and Genomes (KEGG) annotation of the maternal genes was performed using the KEGG web service [36]. Protein sequences in fasta format were blasted in the database.

## 3. Results

### 3.1. Gene Expression Analysis of the Ovary at the Wandering Stage

We previously identified 66 maternal genes in *B*. *mori* [32,33]. At stage 3, a small rise in ecdysteroid titer occurs at the start of wandering, and this is followed by a plateau [24]; then, glycogen synthesis begins in the oocyte, and the follicle cells near the oocyte–nurse cells interface begin to migrate centripetally between the oocyte and the nurse cells [23]. The results of the transcriptional analysis of the 66 maternal genes in the ovary of the wandering silkworm showed transcriptional signals for all the genes (Figure 1 and Appendix A). This is contrary to their low expression in the ovary of the day-3 fifth instar [32]. This result implies the start of the accumulation of maternal mRNAs during stage 3 of the oogenesis developmental period of previtellogenesis, which takes place in the larval period. KEGG ontology assignments of the 66 maternal genes were performed (Appendix A). These genes were mainly found to be involved in genetic information processing and environmental information processing (Appendix A). The genes *eIF4AIII*, *Bin1*, *Pabn2*, *Csp(DnaJ-7)*, *wbl*, *me31B*, and *Mat89Ba* were involved in genetic information processing and were further analyzed. *eIF4AIII*, *Bin1*, and *Pabn2* were involved in the mRNA surveillance pathway, and *Csp(DnaJ-7)*, *wbl*, and *me31B* were involved in RNA degradation (Appendix A).

### 3.2. Expression Profile of Each Gene during Stages 3–5 of Oogenesis

From stages 3 to 5 of silkworm oogenesis, the ovarioles emerged gradually via elongation into the abdominal cavity. The oocyte grew rapidly and was found to contain small protein yolk spheres from stage 4 (Appendix A). To identify the expression profile of each gene during stages 3–5, six time-series samples of ovary or ovariole were collected from the wandering larvae, pre-pupae, day-0 pupae, day-1 pupae, day-2 pupae, and day-3 pupae and were analyzed by RT-qPCR. The thresholds for the up- and downregulation of fold change (≥1.5 and ≤0.67) were found to be differentially reached in the ovarioles of the five other developmental stages in comparison with the wandering stage. In total, the temporal control of the expressional profiles gave rise to three different trends during stages 3–5 (Figure 2, Figure 3, Figure 4 and Table 1). The transcript levels of 18 maternal genes were increased (Figure 2 and Table 1). In contrast, the transcripts of nine genes were significantly decreased (Figure 3 and Table 1). Meanwhile, the transcript levels of 39 genes showed no change (Figure 4 and Table 1). A small rise in ecdysteroid titer occurs at the start of wandering, and this is followed by a plateau. The next day, the titer starts to increase gradually and elevates steeply to form a peak 1 day later [24]. After this peak, the ecdysteroid titer decreases rapidly to a very low level, but then increases again after pupation and reaches a very high titer (forming the second peak) 2 days after pupation [24]. Between the two peaks of the ecdysteroid titer, the silkworm gradually commences larval–pupal metamorphosis, and the organs of the silkworm are reconstructed; specifically, the oocyte grows rapidly. The developmental expression profiles of maternal genes may be closely related to ecdysteroid fluctuation during larval–pupal and the first half of pupal–adult development.

### 3.3. 20-Hydroxyecdysone Regulates the Expression of Select Genes

To confirm the regulatory function of 20E in the expression of maternal genes, an experiment in which exogenous 20E was injected into the 20E-depleted abdomens was performed. The level of ecdysteroid titer is very low in day-0 pupae [24]. When a day-0 pupa is ligated between the thorax and the abdomen, it is found that the abdomen contains no sources of the ecdysteroid that is synthesized and released by the prothoracic glands, which are entwined in pairs in the tracheal bush of the first spiracle in the thorax [37]. This allowed us to examine the effect of 20E. The injection of 20E into 20E-depleted abdomens initially caused significant morphological changes in the ovarian structure, and while the ovarioles developed normally, the development of the ovaries in control individuals was arrested (Appendix A). Our ligation experiments suggest that 20E plays a central role in silkworm oogenesis. We therefore examined the gene expression profiles after 24 and 48 h of 20E treatment. The transcripts of 17 genes (*me31B*, *lok*, *vri*, *CycB*, *aub*, *BAEE*, *PPAE*, *Sod2*, *Smg*, *eIF4AIII*, *rod*, *babo*, *Chc*, *wbl*, *ndl*(*osp*), *pip*, and *tld*) were increased, while the transcripts of 5 genes (*proPPAE*, *PAH*, *esc*, *Nelf-E*, and *Nelf-A*) were decreased, and the transcripts of another 5 genes (*h*, *dpp*, *Egfr*, *Su*(*var*)*205*, and *sog*) showed no change (Figure 5). This indicates that the transcript levels of the maternal genes were regulated by 20E. They may have regulatory function during early silkworm oogenesis.

## 4. Discussion

In insects, ecdysteroids play an important role in molting and metamorphosis. It has been established that insect eggs (including those of the silkworm) contain various ecdysteroids, and the amounts of these ecdysteroids fluctuate during embryonic development [38,39]. Ecdysteroids are essential for embryonic development [40,41] and are synthesized de novo, while free ecdysteroids participate in morphogenesis at an early embryonic stage when the prothoracic glands have not yet differentiated. Thus, ecdysteroids in insect eggs may be of maternal origin [42,43]. Moreover, the increase in 20E titer in young pupae can trigger the first major transition from previtellogenesis to vitellogenesis in silkworm oogenesis [25,26,27]. Meanwhile, the development of animal embryos is initially directed and controlled by the maternal gene products loaded into the egg during oogenesis [1,2]. We previously successfully identified 66 maternal genes in the silkworm [32,33]. In the present study, we first investigated the expression characteristics of the 66 maternal genes in the ovary of the wandering silkworm, and then examined the developmental expression profiles of each gene in the ovary or ovariole from larvae to day-3 pupae, finally confirming the role of 20E in regulating maternal gene expression.

The maternal gene products that are loaded into the egg drive the earliest stages of development, until the zygotic genome can be transcribed. This tightly regulated process is known as the MZT and has been well-studied within model species [2,7,44,45,46]. Nevertheless, we have little knowledge of the time required for this process to begin or of the dynamic landscape of maternal mRNAs during oogenesis in insects. The 66 maternal genes’ transcripts were detected in the ovary of the wandering silkworm. This is contrary to their low expression levels found in the ovaries of day-3 fifth instar insects in our previous study [32]. The wandering silkworm period is stage 3 of oogenesis, at which point the oocyte–nurse cell clusters are arranged in single file in the ovariole, and the follicle cells begin to migrate centripetally between the oocyte and the nurse cells. Glycogen synthesis begins in the oocyte [23]. This indicates the start of the accumulation of maternal mRNAs during stage 3 of the oogenesis developmental period of previtellogenesis in larvae. This obviously differs from the hypothesis that the genes expressed continually during choriogenesis are likely to be maternal genes, because at this stage, the follicle cells gradually experience apoptosis, and the oocyte is uniquely reserved in the mature egg [31,43].

The ovarioles were still enclosed in a protective capsule in day-0 pupae and 20E-depleted abdomens. The protective capsule ruptured, and the ovarioles gradually emerged into the abdominal cavity, shortly after the injection of 20E. In the 20E-depleted abdomens, the ovarioles were still enclosed in a protective capsule following the injection of juvenile hormone (JH) (data not shown). Ovarian development is induced by the injection of tebufenozide into 20E-depleted abdomens, while this is arrested at the stage of mid-vitellogenesis [47]. Ovarian development can be induced secondarily in 20E-depleted abdomens via a single injection of 20E, indicating that 20E plays a central role in silkworm oogenesis and is essential for the initial transition from previtellogenesis to vitellogenesis. In addition, our data show that the dynamic landscape of the 18 genes (incremental expression) and the 9 genes (decremental expression) cohere with the 20E titer changes, along with two peaks during stages 3–5 of silkworm oogenesis. Indeed, we found that the transcripts of 22 genes were regulated in the 20E-depleted abdomens by a single injection of 20E. This finding suggests that 20E plays a central role and regulates the transcription of maternal genes in stages 3–5 of silkworm oogenesis.

*eIF4AIII* (ATP-dependent RNA helicase) is involved in the mRNA surveillance pathway, and *me31B* is involved in RNA degradation, as determined by the KEGG ontology assignments. *eIF4AIII* is essential to the translation of nuclear cap-binding complex (CBC)-associated mRNAs [48,49]. *Me31B* mediates the translational silencing of maternal mRNAs and is an essential regulatory mechanism during early *Drosophila* oogenesis [50,51]. The transcripts of *eIF4AIII* and *me31B* were increased, which may indicate their essential regulatory action during early silkworm oogenesis.

Genes performing essential housekeeping functions are required at all stages of animal development and are transcribed by both the mother and the zygote [7]. The maternal and zygotic genomes are coordinated during early embryonic development to such a degree that the transcript levels of these genes remain relatively constant [7,32], despite the transition between the genomes of origin of these transcripts [7]. The stable mRNAs of 39 maternal genes perform essential housekeeping functions required during stages 3–5 of silkworm oogenesis.

## 5. Conclusions

In summary, this study has taken a much-needed first step towards determining the relationship between 20E and maternal genes during stages 3–5 of silkworm oogenesis. Our findings reveal the role of 20E in regulating the transcription of maternal genes and, accordingly, elucidate the dynamic landscape of gene regulation. Our data also demonstrate the start of the accumulation of maternal mRNAs during stage 3 of the oogenesis developmental period of previtellogenesis in larvae. Our findings expand the understanding of insect oogenesis and provide a perspective on embryo development in *B. mori*.

## Figures and Tables

**Figure 1 insects-12-00969-f001:**
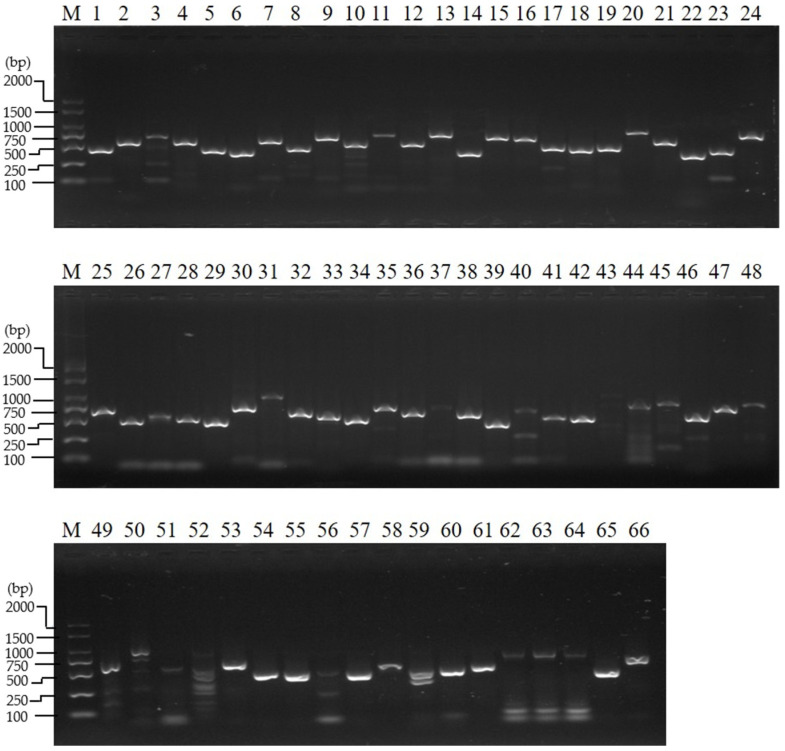
Expression patterns of maternal genes in the ovary of the wandering silkworm by reverse transcription (RT) PCR. M: DL2000 Plus DNA Maker; 1: *me31B*; 2: *lok*; 3: *vri*; 4: *Egfr*; 5: *Su*(*var*) *205*; 6: *Hp1b-l*; 7: *spz*; 8: *tkv*; 9: *CycB*; 10: *proPPAE*; 11: *asp*; 12: *PAH*; 13: *aub*; 14: *Csp*(*DnaJ-7*); 15: *SPE*; 16: *BAEE*; 17: *PPAE*; 18: *Sod2*; 19: *esc*; 20: *Src42A*; 21: *Smg*; 22: *Eif-4a*; 23: *eIF4AIII*; 24: *rod*; 25: *vfl*; 26: *bai*; 27: *Nelf-E*; 28: *Pabn2*; 29: *Bin1*; 30: *tud*; 31: *Moe*; 32: *Sel*(*cnpy1*); 33: *Hip14*(*ZDHHC17*); 34: *mamo*; 35: *sax*; 36: *babo*; 37: *h*; 38: *Chc*; 39: *Snap25*; 40: *SPE-like*; 41: *Src64B*; 42: *wbl*; 43: *Mat89Ba*; 44: *Dif*; 45: *ndl*(*osp*); 46: *Nelf-A*; 47: *tld*; 48: *proSP7*; 49: *gammaTub*; 50: *Th*; 51: *pie*; 52: *gro*; 53: *hb*; 54: *pip*; 55: *spoon*(*AKAP1*); 56: *snk*; 57: *Btk29A*; 58: *dpp*; 59: *Msp300*(*nesprin-1*); 60: *KCNQ*; 61: *shot*; 62: *sog*; 63: *Pc*; 64: *Dst*; 65: *TPH1*; 66: *glo*(*hnRNPF*).

**Figure 2 insects-12-00969-f002:**
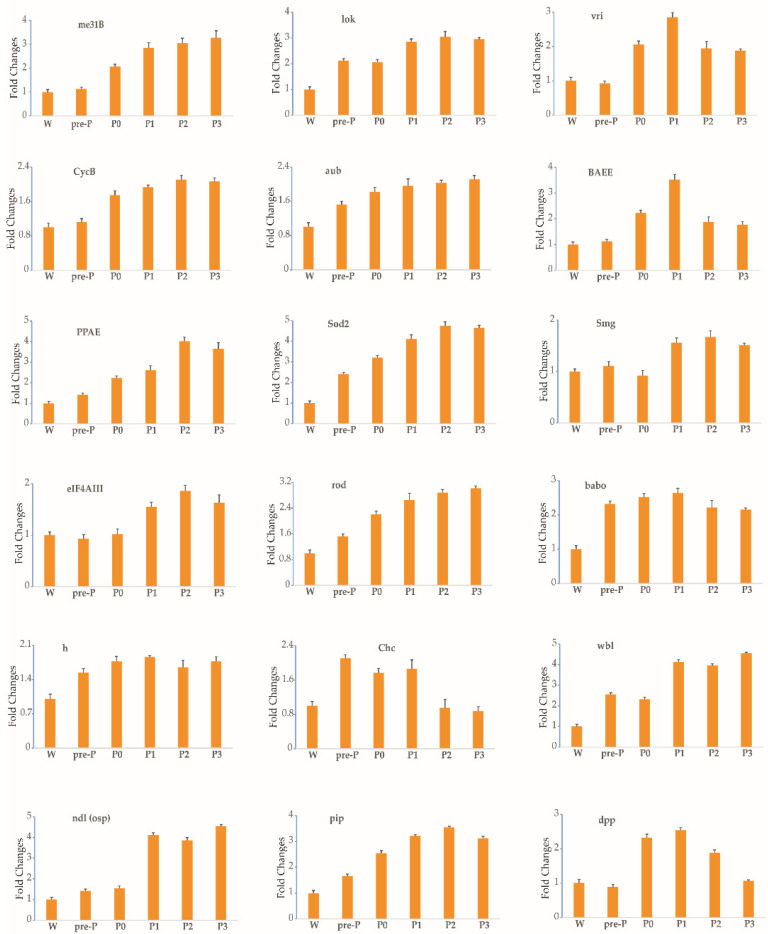
The expression profiles of 18 maternal genes trended upward, as determined by RT-qPCR during stages 3–5 of silkworm oogenesis. Six time-series samples of ovary or ovariole were collected. Each time point was assessed thrice. *TIF-4A* was used as the internal control, and the relative quantities for each gene at the wandering stage were set to 1. Bars indicate the standard deviation. W, wandering larvae; pre-P, pre-pupae; P0, P1, P2 and P3, day-0, day-1, day-2, and day-3 pupae.

**Figure 3 insects-12-00969-f003:**
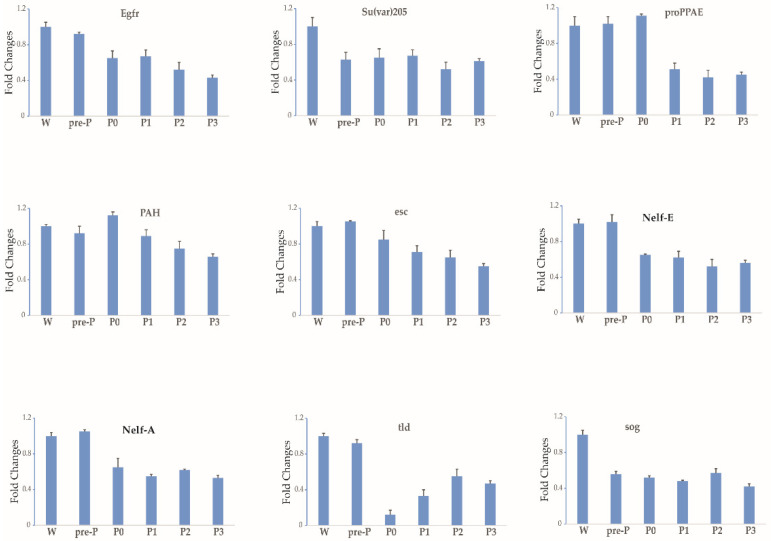
The expression profiles of nine maternal genes trended downward, as determined by RT-qPCR during stages 3–5 of silkworm oogenesis. Six time-series samples of ovaries or ovarioles were collected. Each time point was assessed thrice. *TIF-4A* was used as the internal control, and the relative quantities for each gene at the wandering stage were set to 1. Bars indicate the standard deviation. W, wandering larvae; pre-P, pre-pupae; P0, P1, P2 and P3, day-0, day-1, day-2, and day-3 pupae.

**Figure 4 insects-12-00969-f004:**
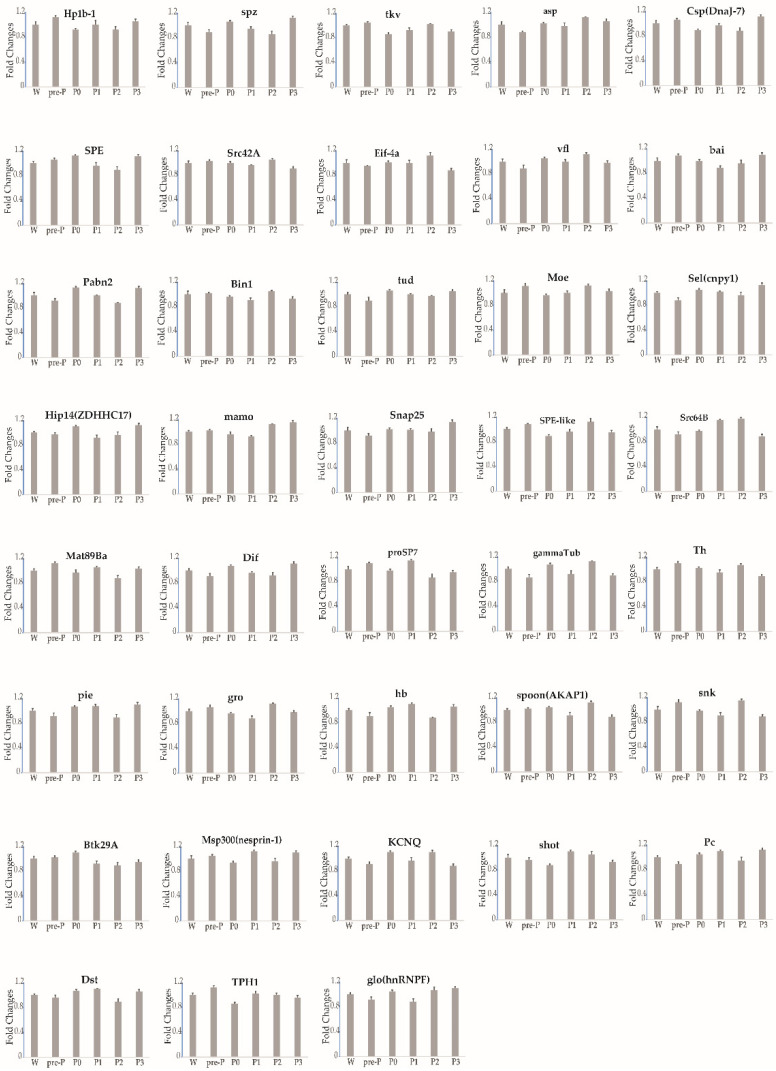
The expression profiles of 39 maternal genes showed a flattened trend, as determined by RT-qPCR during stages 3–5 of silkworm oogenesis. Six time-series samples of ovary or ovariole were collected. Each time point was assessed thrice. *TIF-4A* was used as the internal control, and the relative quantities for each gene at the wandering stage were set to 1. Bars indicate standard deviation. W, wandering larvae; pre-P, pre-pupae; P0, P1, P2 and P3, day-0, day-1, day-2, and day-3 pupae.

**Figure 5 insects-12-00969-f005:**
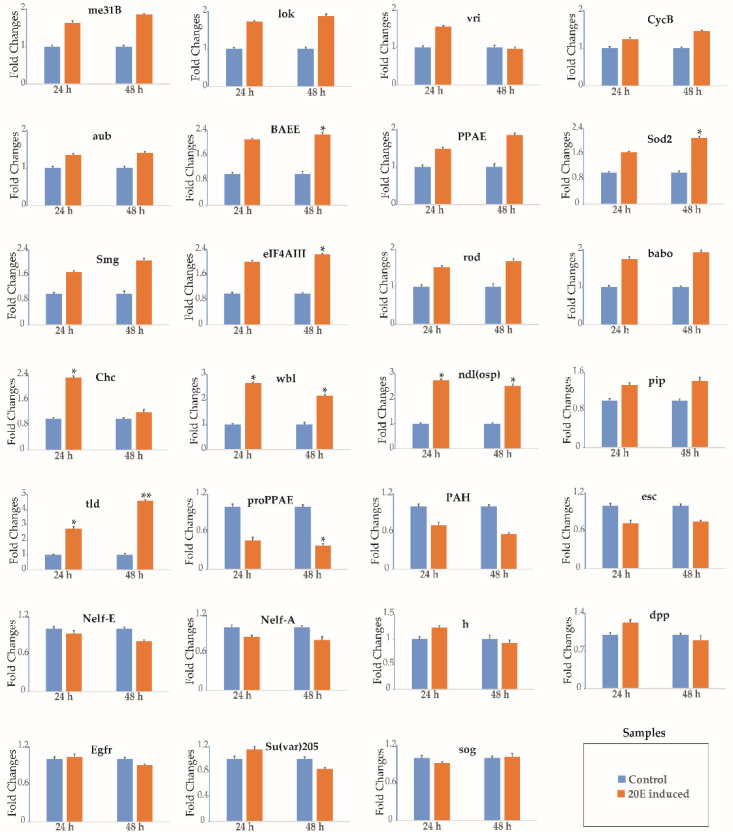
20-hydroxyecdysone (20E) regulation of the expression of select genes determined by RT-qPCR. The isolated abdomens of the pupae (20E-depleted abdomens) were injected with 5 μL of a 20E (10.0 μg) solution or 5 μL of sterile distilled water (control). The samples of ovary and ovariole were collected at 24 h and 48 h after 20E treatment. Each time point was assessed thrice. *TIF-4A* was used as the internal control. Bars indicate the standard deviation. Significant analysis: * *p* < 0.05 or ** *p* < 0.01.

**Table 1 insects-12-00969-t001:** Summary of the expression profile of maternal genes during stages 3–5 of oogenesis.

	No. of Maternal Genes	Name of Maternal Genes
Upward trend	18	*me31B, lok, vri, CycB, aub, BAEE, PPAE, Sod2, Smg, eIF4AIII, rod, babo, h, Chc, wbl, ndl(osp), pip, dpp*
Downward trend	9	*Egfr, Su(var)205, proPPAE, PAH, esc, Nelf-E, Nelf-A, tld, sog*
Flattening trend	39	*Pabn2, Eif-4a, Bin1, bai, tud, gammaTub, Mat89Ba, Pc, Btk29A, Src64B, shot, spoon(AKAP1), Msp300(nesprin-1), TPH1, vfl, KCNQ, Sel(cnpy1), Hip14(ZDHHC17), Hp1b-l, spz, mamo, tkv, hb, snk, asp, proSP7, glo(hnRNPF), Csp(DnaJ-7), pie, SPE, sax, gro, SPE-like, Dif, Th, Src42A, Dst, Snap25, Moe*

## Data Availability

Data are contained within the article and Appendix A.

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
