# Peer review of "Hemolymph Ecdysteroid Titer Affects Maternal mRNAs during Bombyx mori Oogenesis"

_insects, 2021, doi:10.3390/insects12110969_

Round 1

Reviewer 1 Report

Maternal genes play important roles during embryonic development. However, little is known about the dynamic changes of maternal mRNAs in silkworm oogenesis. In this study, the authors determined the developmental expression profile of 66 maternal genes from stages 3 to 5 of silkworm oogenesis and confirmed the role of 20E in regulating maternal gene expression. This article provides important clues for us to further understand the role of maternal genes in embryonic development. I have several comments that the authors need to answer/implement and then the paper can be recommend for publication.

These figures are not clear enough. It is recommended to provide high-resolution image format such as TIFF format.

Figure 1, the authors should mark the size of the DNA marker on the left side of the figure, and write the names represented by the numbers 1-66 in the legend, such as 1-me31B, 2-lok, 3-vri

In the results, the authors should clearly describe the specific methods to divide genes into three clusters.

Figure 1-3, I don't know which two data are used to compare the significance and indicated by the asterisk?

Figure 2-5, the font in Figure 2-5 is too small to see clear.

There are no visible asterisks in Figure 5.

Please write the full names of W, P, pre-P, P0, P1, P2, and P5 in the figure legends.

I suggested that the authors provide a supplementary table to annotate the functions of the genes involved in this article.

Section 3.3, line 205-209, the authors can analyze the affiliation between these genes and the three clusters.

The authors can try to conduct in-depth analysis to see whether there are certain rules between these maternal genes by using KEGG or STRING tools, and add more discussion about the gene function.

Author Response

Response to Reviewer 1 Comments

Point 1: Maternal genes play important roles during embryonic development. However, little is known about the dynamic changes of maternal mRNAs in silkworm oogenesis. In this study, the authors determined the developmental expression profile of 66 maternal genes from stages 3 to 5 of silkworm oogenesis and confirmed the role of 20E in regulating maternal gene expression. This article provides important clues for us to further understand the role of maternal genes in embryonic development. I have several comments that the authors need to answer/implement and then the paper can be recommend for publication.

Response 1: Thanks for the reviewer’s good evaluation and kind suggestion. Due to your suggestion, (â…°): our manuscript has been revised in MDPI for English editing by selecting specialist editing (English editing ID: English-35452); (â…±): the revisions are highlighted using the "Red Font" and "Track Changes" function in Microsoft Word.

Point 2: Figure 1, the authors should mark the size of the DNA marker on the left side of the figure, and write the names represented by the numbers 1-66 in the legend, such as 1-me31B, 2-lok, 3-vri.

Response 2: Thanks for the reviewer’s good evaluation. According to your suggestion, we have made the adjustment in the revision.

Point 3: In the results, the authors should clearly describe the specific methods to divide genes into three clusters.

Response 3: Thanks for the reviewer’s kind suggestion. In our previous manuscript, our description was inappropriate. We're not using the expression approach of ‘cluster’. We have made changes and adjustments that can be found in Line 166-169.

Point 4: Figure 1-3, I don't know which two data are used to compare the significance and indicated by the asterisk?

Response 4: Thanks for the reviewer’s good evaluation. Due to your evaluation, in the revision, (â…°): we have already added “The thresholds for the up- and downregulation of fold change (≥ 1.5 and ≤ 0.67) were found to be differentially expressed in the ovarioles of the five other developmental stages in comparison with the wandering stage.” in Line 166-169; (â…±): we also have already added “the relative quantities for each gene at the wandering stage have been set to 1” in the notes of Figure 2-4; (â…²): basing on your question, we have learned that significance analysis is not necessary in developmental dynamic expression analysis. Therefore, we made changes in the revised manuscript. The advices from you have provided with great helps to us in the current work and will improve our level of scientific research in the future work. Thank you very much.

Point 5: Figure 2-5, the font in Figure 2-5 is too small to see clear. There are no visible asterisks in Figure 5.

Response 5: Thanks for the reviewer’s good evaluation and kind suggestion. We have enlarged the font in the image and increased the resolution. The advices from you have provided with great helps to us in the current work. Thank you very much.

Point 6: Please write the full names of W, P, pre-P, P0, P1, P2, and P3 in the figure legends.

Response 6: Thanks for the reviewer’s kind suggestion. We have added their full names in the figure legends.

Point 7: I suggested that the authors provide a supplementary table to annotate the functions of the genes involved in this article.

Response 7: Thanks for the reviewer’s good evaluation and kind suggestion. We have provided the KEEG functional annotation as a supplementary table S2 in the revision. Thank you very much for your careful reviews in our work.

Point 8: Section 3.3, line 205-209, the authors can analyze the affiliation between these genes and the three clusters.

Response 8: Thanks for the reviewer’s good evaluation. According to your suggestion, we have made the adjustment in the revision. Thank you very much for your careful reviews in our work.

Point 9: The authors can try to conduct in-depth analysis to see whether there are certain rules between these maternal genes by using KEGG or STRING tools, and add more discussion about the gene function.

Response 9: Thanks for the reviewer’s good evaluation. In the revision, we have already done the KEGG annotation of the maternal genes. We have made changes and adjustments that can be found in Line 127-130 in ‘Materials and Methods’ and in Line 276-282 in ‘Discussion’.

Reviewer 2 Report

Comments:

insects-1388458 Profiling the Dynamic Landscape of Maternal mRNAs During Bombyx mori Oogenesis Stages 3-5 Crossing Two Peaks of Hemolymph Ecdysteroid Titer

Maternal genes and ecdysteroids are critical for embryonic development. The silkworm is a good model for the study of maternal genes due to its clearly developmental stage of the oogenesis and long embryonic development. In this manuscript, the authors want to explore the relationship between maternal genes and ecdysteroids. They detected the gene expression pattern of 66 maternal genes at the wandering stage. They found that the transcripts of maternal genes in clusters 1 and 2 were regulated by 20-hydroxyecdysone. However, this manuscript has no new insights currently. The 66 genes tested in the manuscript were also discovered by the previous report. And this manuscript is difficult to understand and read, especially some long sentences. There are many grammatical errors. The manuscript should be modified and improved by the native English professionals.

Title:

Line 1-4 ‘Profiling the Dynamic Landscape of Maternal mRNAs During Bombyx mori Oogenesis Stages 3-5 Crossing Two Peaks of Hemolymph Ecdysteroid Titer’

The title is too long and unclear meaning, the optional one is ‘Hemolymph Ecdysteroid Titer affect the Maternal mRNAs During Bombyx mori Oogenesis’

Authors information:

Personal reminder: It is not common and recommended to use private mailboxes for academic submissions.

Simple Summary:

Line 13: ‘joint relation ’, better use ‘relationship’

Line 15: ‘determined ’, better use ‘detected’

Abstract:

The authors need rewrite the abstract, 1. to make sure what they want to focus on; 2. provide more clearly and readable sentences, avoid use the longer sentences. For examples:

Line 20-22: It’s not clear what this sentence mean. ‘The durations of the stages 3-5 of oogenesis are the silkworms that proceed with larval– pupal metamorphosis and the first half of pupal–adult development crossing two peaks of hemolymph ecdysteroid titer.’ Maybe the author want to say ‘The silkworm larval– pupal metamorphosis and the first half of pupal–adult development happened at the time of oogenesis stages 3-5, which crossing two peaks of hemolymph ecdysteroid titer.’

‘Nevertheless, we have little knowledge of how the pools of the dynamic landscape of maternal mRNAs in silkworm oogenesis.’ difficult to relate to the above sentences.

Line 26, ‘expression characteristics’, better use ‘expression patterns’

Line 26, ‘in ovary of wandering silkworm’, correct is ‘in the ovary at wandering stage’

Line 28, ‘in ovary or ovariole of silkworm from stages 3 to 5 of oogenesis.’, better use ‘ by using ovary or ovariole from stages 3 to 5 of oogenesis.’

Line 30, ‘a single injection of μg quantities’, unclear or wrong description.

Introduction:

The authors should provide more detail information about the maternal genes or proteins in the first paragraph. Which genes and protein were identified in the model organisms, and what their functions in the development.

Line 47-48: ‘Many early developmental processes are highly conserved over evolutionary time and are critical for organism survival’ better use ‘Many early developmental processes are highly evolutionarily conserved and are critical for organism survival’

The second paragraph didn’t relative to this paper, the authors only provided a descriptive process of oogenesis, but did not clarify how these processes are related to maternal inheritance, or what effect of ecdysteroid titer on oogenesis.

Line 49-51, ‘The domestic silkworm, Bombyx mori (B. mori), is not only a domestic insect of its economic relevance to silk production, but also a model insect being completely domesticated of Lepidoptera, which is widely used in basic and applied research’ better use ‘The domestic silkworm, Bombyx mori (B. mori), is not only an economic insect for its silk production, but also as a model for Lepidoptera insect. Therefore, it is widely used in basic and applied research’.

Line 87-90, long sentence, unreadable.

Materials and Methods

Line 105, ‘Larvae of B. mori (Dazao) larvae were reared under standard conditions’ should be ‘B. mori (Dazao) larvae were reared under standard conditions’

Line 117-120, ‘The wandering larvae, pre-pupae, day-0 pupae, day-1 pupae, day-2 pupae, day-3 pupae, 20E-depleted abdomens after injection of 20E 24 and 48h, and the corresponding control isolated abdomens were anaesthetized on ice for 10 min, and the ovaries or ovarioles were collected.’ difficult for understand. Maybe the authors want to say ‘The ovaries or ovarioles were collected from the wandering larvae, pre-pupae, day-0 pupae, day-1 pupae, day-2 pupae, day-3 pupae, which were injected with 20E at 24h and 48h after depleting abdomens. The dissection was conducting on the ice for anaesthetizing’

Results

The gene expression data from the whole ovary, how does the authors distinguish whether these maternal mRNAs come from egg cells or other cells in ovarian tissue? From FigS1, only the day-1 pupa stage can isolate egg cells. Therefore, the genes expression profile made by the authors cannot truly reflect the expression of maternal mRNA. In the model organism Drosophila, the dynamic expression of different genes in a certain cell can be observed by immunofluorescence.

The author should compare to their previous report at the day-3 fifth instar, why chose those days? what’s different? I couldn’t see the 43 express in the Fig1. And gene 3, 10, 17, 50, 52, 56, 59 have multiple bands, which band is correct? The authors need explain these results and give more clearly data.

Line 135, ‘Expression Analysis in 0vary of the Wandering Larvae’ better use ‘Gene Expression Analysis of the Ovary at Wandering Stage’

Line 158, ‘were collected at the wandering’ should be ‘were collected from the wandering’

Significant differences: in the Fig2 and 3, why the authors used difference comparison here, and who compares it with whom? Figure 5 can easily understand that the control and treatment groups can be compared. So Figure 2, 3 should not have this difference comparison.

Author Response

Response to Reviewer 3 Comments

Point 1: Maternal genes and ecdysteroids are critical for embryonic development. The silkworm is a good model for the study of maternal genes due to its clearly developmental stage of the oogenesis and long embryonic development. In this manuscript, the authors want to explore the relationship between maternal genes and ecdysteroids. They detected the gene expression pattern of 66 maternal genes at the wandering stage. They found that the transcripts of maternal genes in clusters 1 and 2 were regulated by 20-hydroxyecdysone. However, this manuscript has no new insights currently. The 66 genes tested in the manuscript were also discovered by the previous report. And this manuscript is difficult to understand and read, especially some long sentences. There are many grammatical errors. The manuscript should be modified and improved by the native English professionals.

Response 1: Thanks for the reviewer’s good evaluation and kind suggestion.

Due to your suggestion:

(â…°): we have revised one by one according to your suggestions. We have carefully revised our manuscript and the longer sentences were avoided. The “introduction” was re-organized with clear logic, adding the maternal genes identified in the model organisms, and their functions in the development in the first paragraph;

(â…±): our manuscript has been revised in MDPI for English editing by selecting specialist editing (English editing ID: English-35452);

(â…²): the revisions are highlighted using the "Red Font" and "Track Changes" function in Microsoft Word;

(â…³): although silkworm is as a model of Lepidoptera and widely used in basic and applied research, the study of maternal genes is not as extensive and detailed as it has been in other model organisms such as Drosophila melanogaster, Caenorhabditis elegans, zebrafish, Xenopus laevis, and mice. Only a few maternal genes have been researched;

(â…´): in our previous reports, firstly, we identify maternal-related genes from silkworm unfertilized eggs from five different stages (0 h, 6 h, 12 h, 18 h, 24 h). 0 h is defined as the 15th minutes after most virgin moths spawn. (The title: Identifying potential maternal genes of Bombyx mori using digital gene expression Profiling, PLoS One. 2018 Feb 20;13(2):e0192745. doi: 10.1371/journal.pone.0192745.eCollection2018). Secondly, identifying the maternal genes via orthologous comparison and expression detection in eggs of virgin moths, and their mRNAs degradation patterns during the maternal to zygotic transition (MZT) period in the silkworm. (The title: Expression Analysis of mRNA Decay of Maternal Genes during Bombyx mori Maternal-to-Zygotic Transition, Int J Mol Sci. 2019 Nov 12;20(22):5651. doi: 10.3390/ijms20225651).

Remarkably, the development stages of silkworm oogenesis are clear and the study materials are easy to obtain. In this study, choosing the silkworm oogenesis stages 3-5 to explore the start of the accumulation of maternal mRNAs and the relationship between maternal genes and ecdysteroids during the silkworm oogenesis. These results are a new study with originality, and are also the continuation of our previous work. The advices from you have provided with great helps to us in the current work. Thank you very much for your careful reviews in our work.

Point 2: Line 1-4 ‘Profiling the Dynamic Landscape of Maternal mRNAs During Bombyx mori Oogenesis Stages 3-5 Crossing Two Peaks of Hemolymph Ecdysteroid Titer’. The title is too long and unclear meaning, the optional one is ‘Hemolymph Ecdysteroid Titer affect the Maternal mRNAs During Bombyx mori Oogenesis’.

Response 2: Thanks for the reviewer’s good evaluation and kind suggestion. Due to your suggestion, we used the title that you suggested ‘Hemolymph Ecdysteroid Titer affect the Maternal mRNAs During Bombyx mori Oogenesis’. The advices from you have provided with great helps to us in the current work. Thank you very much.

Point 3: Authors information:

Personal reminder: It is not common and recommended to use private mailboxes for academic submissions.

Response 3: Thanks for the reviewer’s kind suggestion. We used the office mailbox instead of the private one. The advices from you have provided with great helps to us in the current work. Thank you very much.

Point 4: Simple Summary:

Line 13: ‘joint relation ’, better use ‘relationship’

Line 15: ‘determined ’, better use ‘detected’

Response 4: Thanks for the reviewer’s good evaluation. Due to your suggestion, we have revised our manuscript. Thank you very much for your careful reviews in our work.

Point 5: Abstract:

The authors need rewrite the abstract, 1. to make sure what they want to focus on; 2. provide more clearly and readable sentences, avoid use the longer sentences. For examples: Line 20-22: It’s not clear what this sentence mean. ‘The durations of the stages 3-5 of oogenesis are the silkworms that proceed with larval– pupal metamorphosis and the first half of pupal–adult development crossing two peaks of hemolymph ecdysteroid titer.’ Maybe the author want to say ‘The silkworm larval– pupal metamorphosis and the first half of pupal–adult development happened at the time of oogenesis stages 3-5, which crossing two peaks of hemolymph ecdysteroid titer.’

Response 5: Thanks for the reviewer’s good evaluation. Due to your suggestion, we have revised our manuscript. Thank you very much for your careful reviews in our work.

Point 6: ‘Nevertheless, we have little knowledge of how the pools of the dynamic landscape of maternal mRNAs in silkworm oogenesis.’ difficult to relate to the above sentences.

Response 6: Thanks for the reviewer’s kind suggestion. This sentence is very abrupt and it has been deleted by us in the revision.

Point 7:

Line 26, ‘expression characteristics’, better use ‘expression patterns’

Line 26, ‘in ovary of wandering silkworm’, correct is ‘in the ovary at wandering stage’

Line 28, ‘in ovary or ovariole of silkworm from stages 3 to 5 of oogenesis.’, better use ‘ by using ovary or ovariole from stages 3 to 5 of oogenesis.’

Line 30, ‘a single injection of μg quantities’, unclear or wrong description.

Response 7: Thanks for the reviewer’s kind suggestion. We have revised the manuscript according to your suggestion. Thank you very much for your careful reviews in our work.

Point 8: The authors should provide more detail information about the maternal genes or proteins in the first paragraph. Which genes and protein were identified in the model organisms, and what their functions in the development.

Response 8: Thanks for the reviewer’s good evaluation. We have carefully revised our manuscript. The “introduction” was re-organized with clear logic, adding the maternal genes identified in the model organisms, and their functions in the development in the first paragraph. They can be found in Line 40-49 in ‘Introduction’.

Point 9: Line 47-48: ‘Many early developmental processes are highly conserved over evolutionary time and are critical for organism survival’ better use ‘Many early developmental processes are highly evolutionarily conserved and are critical for organism survival’.

Response 9: Thanks for the reviewer’s kind suggestion. In our previous manuscript, our description was inappropriate. We have made changes and adjustments according to your suggestion. Thank you very much for your careful reviews in our work.

Point 10: The second paragraph didn’t relative to this paper, the authors only provided a descriptive process of oogenesis, but did not clarify how these processes are related to maternal inheritance, or what effect of ecdysteroid titer on oogenesis.

Response 10: Thanks for the reviewer’s good evaluation. We have carefully revised our manuscript. Due to your suggestion, we have made changes and adjustments in the revision. The advices from you have provided with great helps to us in the current work and will improve our level of scientific research in the future work. Thank you very much.

Point 11: Line 49-51, ‘The domestic silkworm, Bombyx mori (B. mori), is not only a domestic insect of its economic relevance to silk production, but also a model insect being completely domesticated of Lepidoptera, which is widely used in basic and applied research’ better use ‘The domestic silkworm, Bombyx mori (B. mori), is not only an economic insect for its silk production, but also as a model for Lepidoptera insect. Therefore, it is widely used in basic and applied research’.

Line 87-90, long sentence, unreadable.

Response 11: Thanks for the reviewer’s good evaluation. We have carefully revised our manuscript and the longer sentences were avoided. Due to your suggestion, we have made changes and adjustments in the revision. The advices from you have provided with great helps to us in the current work. Thank you very much.

Point 12: Materials and Methods

Line 105, ‘Larvae of B. mori (Dazao) larvae were reared under standard conditions’ should be ‘B. mori (Dazao) larvae were reared under standard conditions’

Line 117-120, ‘The wandering larvae, pre-pupae, day-0 pupae, day-1 pupae, day-2 pupae, day-3 pupae, 20E-depleted abdomens after injection of 20E 24 and 48h, and the corresponding control isolated abdomens were anaesthetized on ice for 10 min, and the ovaries or ovarioles were collected.’ difficult for understand. Maybe the authors want to say ‘The ovaries or ovarioles were collected from the wandering larvae, pre-pupae, day-0 pupae, day-1 pupae, day-2 pupae, day-3 pupae, which were injected with 20E at 24h and 48h after depleting abdomens. The dissection was conducting on the ice for anaesthetizing’.

Response 12: Thanks for the reviewer’s good evaluation. Due to your suggestion, we have made changes and adjustments in the revision in “Materials and Methods”. The advices from you have provided with great helps to us in the current work. Thank you very much.

Point 13: The gene expression data from the whole ovary, how does the authors distinguish whether these maternal mRNAs come from egg cells or other cells in ovarian tissue? From FigS1, only the day-1 pupa stage can isolate egg cells. Therefore, the genes expression profile made by the authors cannot truly reflect the expression of maternal mRNA. In the model organism Drosophila, the dynamic expression of different genes in a certain cell can be observed by immunofluorescence.

Response 13: Thanks for the reviewer’s good evaluation.

There is ovarian membranes outside the ovary of bombyx mori. Internal differentiation is not obvious at the time of hatching. Three days later, the ovary is composed of four loculi. After the day-3 fifth instar, the loculi begin to develop. The loculi develop into tubules, and ovarian tubes circulate irregularly in the ovary at the wandering stage. The paired ovaries are each composed of four ovarian tubes, each of which contains a chain of follicles in progressively advanced stages of development. Each follicle is made up of an oocyte and seven nurse cells surrounded by a single layer of follicular epithelium derived from mesodermal cells. The oocyte and its accompanying nurse cells are third-generation descendants of a final oogonium. It is difficult to distinguish whether the maternal mRNAs come from oocyte or nurse cell in earlier oogenesis of silkworm. Undeniably, it is perfect in combining with the immunofluorescence results. Meanwhile, the ovarioles emerge gradually via elongation into the abdominal cavity at day- pupae, which is part of the oogenesis developmental periods of vitellogenesis and choriogenesis. In addition, the follicles in each ovarian tube at day-8 pupae can be subdivided into the three distinct developmental periods of previtellogenesis, vitel-logenesis and choriogenesis. The method of immunofluorescence can be used well in the middle and late stages of silkworm oogenesis. The advices from you have provided with great helps to us in the current work and will improve our level of scientific research in the future work. Thank you very much.

Point 14: The author should compare to their previous report at the day-3 fifth instar, why chose those days? what’s different? I couldn’t see the 43 express in the Fig1. And gene 3, 10, 17, 50, 52, 56, 59 have multiple bands, which band is correct? The authors need explain these results and give more clearly data.

Response 14: Thanks for the reviewer’s kind suggestion.

The silkworm feeds and grows quickly in the fifth larval period. Day 3 of the fifth instar of the silkworm is the boundary for whole larval development stage. The fifth instar of silkworm larva feeds and grows quickly before this time, but afterward, the silkworm hemolymph proteins gradually change and silkworms to synthesize massively silk proteins in the silk gland. Therefore, studying this time point will enrich the expression patterns and help with further understanding of the functions of maternal genes in dierent developmental stages.

Meanwhile, the development of the follicles is divided into 12 different stages according to various morphological criteria. For the developmental stages of the follicles, the stage 1 is on the day-1 fourth instar, the stage 2 is on the day-2 fourth instar, the stage 3 is on the day-7 fifth instar (the start of wandering), the stage 4 is on the day-1 pupa, and the stage 5 is on the day-3 pupa. Moreover, two peaks of hemolymph ecdysteroid titer accompany the silkworm oogenesis stages 3 to 5. Glycogen synthesis begins in the oocyte at stage 3, to stage 5, the oocyte accounts for one-half of follicular volume and its nucleus lies in an anterior dorsal position of the ooplasm. The oocyte grows rapidly after stage 5. Therefore, we chose the stages 3 to 5 in this study.

The band (43) is weak, and transcription levels of Mat89Ba (43) can be detected in ovary of wandering silkworm by reverse transcription-quantitative PCR (RT-qPCR). TIF-4A was used as the internal control and the relative quantity for TIF-4A was set to 1.

gene 3, 10, 17, 50, 52, 56, 59 have multiple bands:

52 (gro): has two CDS sequences, one CDS 492bp, another CDS 201bp, the bands of RT-PCR result including the two CDS sequences, but RT-qPCR result including the CDS 492bp.

59 (Msp300(nesprin-1)): has a very large CDS sequence (46,278bp), One part of the sequences was selected to design the primer.

3 (vri), 10 (proPPAE), 17 (PPAE), 50 (Th), and 56 (snk):

each of them has one CDS sequence, the results of blasting in NCBI and SilkDB 3.0 (https://silkdb.bioinfotoolkits.net/main/species-info/-1) database also showing one loaded sequence.

The advices from you have provided with great helps to us in the current work and will improve our level of scientific research in the future work. Thank you very much.

Point 15:

Line 135, ‘Expression Analysis in 0vary of the Wandering Larvae’ better use ‘Gene Expression Analysis of the Ovary at Wandering Stage’

Line 158, ‘were collected at the wandering’ should be ‘were collected from the wandering’.

Response 15: Thanks for the reviewer’s good evaluation. Due to your suggestion, we have made changes and adjustments in the revision.

Point 16:

Significant differences: in the Fig2 and 3, why the authors used difference comparison here, and who compares it with whom? Figure 5 can easily understand that the control and treatment groups can be compared. So Figure 2, 3 should not have this difference comparison.

Response 16: Thanks for the reviewer’s good evaluation. Due to your evaluation, in the revision, (â…°): we have already added “the relative quantities for each gene at the wandering stage have been set to 1” in the notes of Figure 2-4;

(â…±): basing on your question, we have learned that significance analysis is not necessary in developmental dynamic expression analysis. Therefore, we have made changes in the revised manuscript.

Thanks for your kind suggestion and good evaluation. We have revised one by one according to your suggestions. Thank you very much for your careful reviews in our work. The advices from you have provided with great helps to us in the current work. In our future work on silkworm maternal genes, we have obtained the inspirations and ideas from your strategic and constructive suggestions. The methods of experimental study and data analysis that you mentioned (comments on this study) have improved and enriched our understanding of scientific issues. It is our honors for our work getting your guidance and evaluation. Thank you very much.

Reviewer 3 Report

     Maternally provided gene products, namely mRNA and proteins, are essential for the early development of embryos. These molecules are transferred from maternal tissues to oocytes during oogenesis and vitellogenesis. Authors have previously identified 66 maternal genes in the silkworm Bombyx mori. In the present study, they analyzed expression profiles of those genes at a transition period from previtellogenic to vitellogenic periods. They also examined the effect of 20-hydroxyecdysone on their expression. I have several concerns on the manuscript for the publication as follows:

  1. Gene clustering

     According to the expression profiles, the 66 genes were clustered into three clusters. However, this clustering process lacks objective evidences. It seems that cluster 1, 2, and 3 include genes whose expression increased, decreased, and did not change, during the experimental period, respectively. However, some genes in cluster 1 decreased their expression after pupation. At least it is necessary to state the clustering criteria.

  1. Biological significance of the clusters

     The manuscript is highly descriptive. What does the three clusters indicate? How do genes in each cluster contribute on oogenesis or embryogenesis?

  1. Figure 1

     Biological significance of Figure 1 is obscure. According to this data, authors described the accumulation of mRNA at this time points, even though this is inconsistent to their previous data. Because PCR is highly sensitive, it can detect mRNA existing at very low level. Quantitative analyses or other analyses to show the accumulation are necessary.

  1. 20E effect on gene expression

    Genes in cluster 1 and 2, whose expression were fluctuated during the experimental period and so which were concerned to be regulated by ecdysteroids, were analyzed for effect of 20E injection. Several genes (h,dpp, Egfr, Su(var)205, sog, and tld) showed results inconsistent to their expression pattern. How are these results explained?

Other minor concerns are listed below:

  1. Final instar of B. mori

Readers who do not know about B. mori might not know that fifth instar is the final instar. This is better to be stated.

  1. Figures 2-4

Figures are too small to read. Even in the online version, it is hard to read the gene names and stages. The Y-axes are “fold change”, but the standard, namely value 1, is not clear. The method for the statistical analysis should be stated.

  1. Line 135

“Ovary” in the section title looks like starting with “0”.

  1. Line 230

“66 genes maternal genes” should be “66 maternal genes”.

Author Response

Response to Reviewer 5 Comments

Point 1: Maternally provided gene products, namely mRNA and proteins, are essential for the early development of embryos. These molecules are transferred from maternal tissues to oocytes during oogenesis and vitellogenesis. Authors have previously identified 66 maternal genes in the silkworm Bombyx mori. In the present study, they analyzed expression profiles of those genes at a transition period from previtellogenic to vitellogenic periods. They also examined the effect of 20-hydroxyecdysone on their expression.

Response 1: Thanks for the reviewer’s good evaluation and kind suggestion. We have revised one by one according to your suggestions. Due to your suggestion, (â…°): our manuscript has been revised in MDPI for English editing by selecting specialist editing (English editing ID: English-35452); (â…±): the revisions are highlighted using the "Red Font" and "Track Changes" function in Microsoft Word. Thank you very much for your careful reviews in our work.

Point 2: Gene clustering

According to the expression profiles, the 66 genes were clustered into three clusters. However, this clustering process lacks objective evidences. It seems that cluster 1, 2, and 3 include genes whose expression increased, decreased, and did not change, during the experimental period, respectively. However, some genes in cluster 1 decreased their expression after pupation. At least it is necessary to state the clustering criteria.

Response 2: Thanks for the reviewer’s good evaluation. Due to your evaluation, in the revision, (â…°): we have already added “The thresholds for the up- and downregulation of fold change (≥ 1.5 and ≤ 0.67) were found to be differentially expressed in the ovarioles of the five other developmental stages in comparison with the wandering stage.” in Line 166-169; (â…±): we also have already added “the relative quantities for each gene at the wandering stage have been set to 1” in the notes of Figure 2-4; (â…²): basing on your question, we have learned that significance analysis is not necessary in developmental dynamic expression analysis. Therefore, we made changes in the revised manuscript. The advices from you have provided with great helps to us in the current work and will improve our level of scientific research in the future work. Thank you very much.

Point 3: 2.    Biological significance of the clusters

The manuscript is highly descriptive. What does the three clusters indicate? How do genes in each cluster contribute on oogenesis or embryogenesis?

Response 3: Thanks for the reviewer’s kind suggestion. We have provided the KEEG functional annotation as a supplementary table S2 in the revision. We have made changes and adjustments that can be found in Line 127-130 in ‘Materials and Methods’ and in Line 276-282 in ‘Discussion’. Thank you very much for your careful reviews in our work.

Point 4: Fig. 1: Biological significance of Figure 1 is obscure. According to this data, authors described the accumulation of mRNA at this time points, even though this is inconsistent to their previous data. Because PCR is highly sensitive, it can detect mRNA existing at very low level. Quantitative analyses or other analyses to show the accumulation are necessary.

Response 4: Thanks for the reviewer’s good evaluation. In this study, we aimed to explore the start of the accumulation of maternal mRNAs.

In the revision, we added Figure S1:

Expression patterns of maternal genes in ovary of wandering silkworm by reverse transcription-quantitative PCR (RT-qPCR). TIF-4A was used as the internal control and the relative quantity for TIF-4A was set to 1.

The advices from you have provided with great helps to us in the current work and will improve our level of scientific research in the future work. Thank you very much.

Point 5: 4.    20E effect on gene expression20E

Genes in cluster 1 and 2, whose expression were fluctuated during the experimental period and so which were concerned to be regulated by ecdysteroids, were analyzed for effect of 20E injection. Several genes (h,dpp, Egfr, Su(var)205, sog, and tld) showed results inconsistent to their expression pattern. How are these results explained?

Response 5: Thanks for the reviewer’s kind suggestion. In figures 2 and 3, the relative quantities for each gene at the wandering stage have been set to 1.

In figure 5: the change of hemolymph ecdysteroid titer from wandering stage to day-0 pupae: a small rise in ecdysteroid titer occurs upon commencement of wandering, and this is followed by a plateau. The next day, the titer starts to increase gradually, and then elevates steeply to form a peak 1 day later. The isolated abdomens of the pupae are day-0 pupae. The relative quantities for each select gene at the 20E-depleted abdomens of day-0 pupae have been set to 1. The expression level of genes (h,dpp, Egfr, Su(var)205, sog, and tld) has regulated by the first peak of hemolymph ecdysteroid titer in silkworm. Thank you very much for your careful reviews in our work.

Point 6: Final instar of B. mori

Readers who do not know about B. mori might not know that fifth instar is the final instar. This is better to be stated. mori.

Response 6: Thanks for the reviewer’s kind suggestion. According to your suggestion, we have made the adjustment in the revision. Thank you very much for your careful reviews in our work.

Point 7: Figures 2-4

Figures are too small to read. Even in the online version, it is hard to read the gene names and stages. The Y-axes are “fold change”, but the standard, namely value 1, is not clear. The method for the statistical analysis should be stated.

Response 7: Thanks for the reviewer’s good evaluation and kind suggestion. We have enlarged the font in the image and increased the resolution. The advices from you have provided with great helps to us in the current work. Thank you very much.

Point 8: Line 135

Ovary” in the section title looks like starting with “0”.

Line 230

66 genes maternal genes” should be “66 maternal genes”.

Response 8: Thanks for the reviewer’s good evaluation and kind suggestion. According to your suggestion, we have made the adjustment in the revision. Thank you very much for your careful reviews in our work.

Round 2

Reviewer 2 Report

The authors addressed all of my questions. The revised manuscript has been greatly improved, I agree to publish in the current version.